

# Global maps of Forel-Ule index, hue angle and Secchi disk depth derived from twenty-one years of monthly ESA-OC-CCI data

Jaime Pitarch[1,2], Marco Bellacicco[3], Salvatore Marullo[1,3], Hendrik J. van der Woerd[4]

[1]Consiglio Nazionale delle Ricerche (CNR), Istituto di Scienze Marine (ISMAR), Via Fosso del Cavaliere 100, 00133 Rome, Italy
[2]NIOZ Royal Netherlands Institute for Sea Research, Department of Coastal Systems, and Utrecht University, PO Box 59, 1790AB Den Burg, Texel, The Netherlands
[3]Italian National Agency for New Technologies, Energy and Sustainable Economic Development (ENEA), Via E. Fermi 45, 00044 Frascati, Italy
[4]Institute for Environmental Studies (IVM), Water & Climate Risk, Vrije Universiteit Amsterdam, De Boelelaan 1111, 1081HV Amsterdam, The Netherlands

*Correspondence to*: Jaime Pitarch (jaime.pitarchportero@artov.ismar.cnr.it)

**Abstract.** We document the development and public release of a new dataset (1997-2018), consisting of global maps of the Forel-Ule index, hue angle and Secchi disk depth. Source data comes from the European Space Agency (ESA) Ocean Colour (OC) Climate Change Initiative (CCI), which is providing merged multi-sensor data from the mid-resolution sensors in operation at a specific time from 1997 to the present day. Multi-sensor satellite datasets are advantageous tools for ecological studies because they increase the probabilities of cloud-free data over a given region, as data from multiple satellites whose overpass times differ by a few hours are combined. Moreover, data merging from heritage and present satellites can expand the duration of the time series indefinitely, which allows the calculation of significant trends. Additionally, data are remapped consistently and analysis-ready for scientists. Also, the products described in this article have the exclusive advantage of being linkable to in-situ historic observations and thus enabling the construction of very long time series. Monthly data are presented at a spatial resolution of ~4 km at the equator and are available at PANGAEA, https://doi:10.1594/PANGAEA.904266 (Pitarch et al., 2019a). Two smaller and easier to handle test datasets have been produced from the former: a global dataset at 1 degree spatial resolution and another one for the North Atlantic at 0.25 degree resolution.

## 1. Introduction

Single-sensor satellite datasets are not long enough to provide significant evidence of climatic trends that become manifest over fluctuations, so the multi-sensor approach has been fostered since the last decade (IOCCG, 2007). In Europe, the ESA's Climate Change Initiative (CCI) was created to address the United Nations Framework Convention on Climate Change requirements on a systematic monitoring of the global climate system. The Global Climate Observing System (GCOS) defined a number of Essential Climate Variables (ECVs), which are physical, chemical or biological variables that critically contributes to the characterisation of Earth's climate. Ocean Colour is one of the ECV and the OC-CCI is devoted to



producing a time series of consistent measurements encompassing water-leaving radiance in the visible domain, derived chlorophyll and inherent optical properties, based on merged MERIS, Aqua-MODIS, SeaWiFS and VIIRS satellite data (Jackson et al., 2019).

While the chlorophyll concentration has historically been the most studied variable from ocean colour observations, more variables can be derived with confidence that can provide further insights, such as inherent optical properties or the diffuse attenuation coefficient ($K_d$). Algorithm developers can also download the remote-sensing reflectance ($R_{rs}$) at ftp.oceancolor.org, and derive any other geophysical property themselves.

In the last decade, two historic optical variables have received renewed attention: the Secchi disk depth ($z_{SD}$) and the Forel-Ule (FU) colour index. These two are the oldest oceanographic variables that are directly related to ocean colour, recorded for many decades before the advent of satellites (Boyce et al., 2012) and publicly available globally (NOAA, 2013). The Secchi disk is a simple white disk that is lowered from above the water surface and is tracked visually until it goes out of sight. The depth when it

ceases to be visible provides a reading of water transparency, (Wernand, 2010). $z_{SD}$ is influenced by the underwater light attenuation, which in turn depends on the light absorption and scattering caused by the varying concentrations of dissolved and suspended substances in the water. The $z_{SD}$ observation also depends on disk design and environmental factors that need to be understood (Pitarch, 2020). The popularity and simplicity of this practice allowed its generalized use in oceanic surveys during the last

century, although its use declined significantly in the last decades. Currently, the method is experiencing a revival due to recent efforts to derive this quantity from remote sensing data (Lee et al., 2015).
      The FU scale was developed as a set of standard colours that would allow visual water colour indexing (Wernand and Van Der Woerd, 2010). As with $z_{SD}$, the colour of natural waters is also influenced by substances in the water, present in different concentrations, which constitutes the basis for ocean colour

science. The popularity of the FU scale has had a similar history as the Secchi disk, and has evolved from being widely adopted, then mostly abandoned and recently revived, aided by the link to remote-sensing data (Wernand et al., 2013).
      The hue angle is used as an intermediate quantity in the process of deriving FU from any light spectrum (Wernand et al., 2013) and it can be regarded as the continuous companion of the FU scale. Specific

algorithms have been developed for the spectral characteristics of various ocean colour sensors (Van Der Woerd and Wernand, 2015;Van Der Woerd and Wernand, 2018). In terms of in-situ sampling, this quantity cannot be estimated visually but only radiometrically, although Red Green Blue (RGB) images have proven good for this sake, thus enabling the use of digital cameras (Ceccaroni et al., 2020).
      Pitarch et al. (2019b) made an analysis of global seasonal variability of these parameters using the

climatological ESA-OC-CCI data, including some cross-relationships and ranges of variability.
      Despite the demonstrated usefulness of these optical parameters in marine physics and biogeochemistry, no public dataset of them exists so far, so scientists need to calculate them $R_{rs}$ privately, which, other than creating redundant efforts, may be a too high wall to overcome for non-experts. This gap is filled in the present article with the provision of the monthly series from 1997 to 2018 of Secchi disk, hue angle and

Forel-Ule index at ~4 km resolution.



## 2. Methods

### 2.1. Ingested data

Downloaded product is the merged multi-sensor OC-CCI v4.2 remote-sensing reflectance ($R_{rs}$), which is the primary quantity used for ocean colour studies, containing only spectral information related to the colour of the water, while having all interfering factors removed in the atmospheric correction and normalization calculations. $R_{rs}$ is provided at a monthly frequency and projected on a rectangular grid of 2.5 minutes of arc for both latitude and longitude, which corresponds to about 4 km at the equator, and

decreasing poleward. For more specific details, one can read the Product User Guide (Jackson et al., 2019). Every file of Secchi disk, hue angle and Forel-Ule keeps the same format as the corresponding source $R_{rs}$ file. Numeric precision is single (32-bit floating-point values), except time, which is 32-bit signed integer. Every downloaded data file contains a map of estimated bias in $R_{rs}$, which we add to generate the unbiased $R_{rs}$ estimate. This bias compensation, which users need to perform themselves, has

proven to enhance the quality of retrievals of $R_{rs}$-derived geophysical products like optical particle backscattering (Pitarch et al., 2020).

### 2.2. The Forel-Ule and Hue angle algorithms

    The hue angle is a continuous numerical variable that expresses the colour of a light spectrum with a single number. In natural waters, the hue angle ranges from about 40° in brown waters to about 235° in

the deep blue oceanic waters. As the source spectra and colour perception cover the continuous spectral range, some kind of adaptation needs to be done when applying it to satellite data that is provided only at a small number of wavelengths. Full technical details about hue angle determination from $R_{rs}$ are published (Van Der Woerd and Wernand, 2015;Van Der Woerd and Wernand, 2018;Pitarch et al., 2019b). The FU scale was optically characterized as a canonical set of points (x, y) in the CIE space (Novoa et

al., 2013). Because the hue angle is the polar angle of the (x, y) coordinates, the hue angle of each $R_{rs}$ spectrum can be calculated and any measured hue angle can be assigned an FU index by choosing the nearest FU index in hue angle terms. In fact, FU can be seen as a discrete version of the hue angle and can be used to cluster water masses in terms of their colour. However, it was found that the FU scale lacks resolution and dynamic range for the most oligotrophic oceanic waters (Pitarch et al., 2019b). Vast regions

of the ocean remain saturated all year round at FU=1. For this reason, a new lower end FU=0 was proposed in order to introduce more differentiation within the bluest waters. This dataset includes that addition.

### 2.3. The Secchi disk depth algorithm

    We apply the state-of-the-art algorithm for the remote estimation of the Secchi disk depth ($z_{SD}$) by Lee et al. (2015). According to the underlying theory, $z_{SD}$ essentially depends on the spectral minimum of the

diffuse attenuation coefficient ($K_d$), which is analytically modelled as a function of the absorption (a) and optical backscattering coefficient ($b_b$) (Lee et al., 2013). The two latter are retrieved with the QAAv6 algorithm (Lee et al., 2002). Notably, the retrieved $b_b$ is empirically corrected for Raman scattering after Lee et al. (2013). It has been shown that such step improves the match of the retrieved $b_b$ against reference



data (Pitarch et al., 2020). Solar zenith angle, which is also needed for $K_d$, is set equal to zero, to be consistent with the convention used for OC-CCI satellite normalized reflectances.

### 3. Product description

The processing is applied on a pixel basis from every source netCDF $R_{rs}$ file. Therefore, the format of the product file is the same and the spatial and temporal variables are directly copied from the source file. The hue angle, the Forel-Ule index and the Secchi disk depth are added. Content of each file is
summarized in Table 1. Every netCDF file has the naming ESACCI-OC-L3S-MERGED-1M_MONTHLY_4km_GEO_PML_Hue_FU_SD-yyyymm-fv4.0.nc, where yyyy is a four character string for the year and mm is a four character string for the month.

**Table 1 File description**

| Parameter | Description | Dimensions | Size (pixels) |
|-----------|-------------|------------|---------------|
| lat | Geographic latitude, positive northward | degree | 4320x1 |
| lon | Geographic longitude, positive eastward | degree | 8640x1 |
| time | Time since 1970-01-01 00:00:00 | day | 1x1 |
| alfa | Hue angle, according to (Pitarch et al. (2019b) | degree | 8640x4320x1 |
| FU | Extended Forel-Ule index of the remote-sensing reflectance (0 to 21), according to (Pitarch et al. (2019b) | - | 8640x4320x1 |
| z_SD | Secchi disk depth, according to Lee et al. (2015) | m | 8640x4320x1 |

### 4. Validation

Validation of water quality parameters is an ongoing activity, even for consolidated products like chlorophyll and reflectance. In the remote sensing community, validation is mostly known as a comparison to in-situ reference data. Assuming errorless reference data, uncertainties are then due to the atmospheric correction and the applied water quality algorithms. However, in-situ data also contains
uncertainties that are seldom considered.
Monthly OC-CCI products cannot be used for matchup to in-situ data as matchups are intended for instantaneous data, with time coincidence between the satellite and the in-situ readings recommended to be within few hours. Understanding this rule in a broad sense, daily satellite data, also provided within OC-CCI, may be acceptable. Therefore, the uncertainties can be assessed using daily data, knowing that
uncertainties present in the monthly aggregates will be smaller, as the temporal averaging will dampen random noise by a factor equal to the square root of the number of observations at every pixel for every month. This can be seen in the produced monthly images, where speckle noise looks rather low. Unfortunately, this number of observations per pixel in each month is unknown for the monthly OC-CCI products.



### 4.1. Remote-sensing reflectance

The OC-CCI's Product User Guide (PUG) (Jackson et al., 2019) documents uncertainty assessment of satellite OC-CCI $R_{rs}$ by comparing to large in-situ $R_{rs}$ database that merges in-situ $R_{rs}$ from many different sources with very disparate instrumentation and quality standards (Valente et al., 2019) and without an uncertainty budget assessment. Use of this entire database can artificially increase the error statistics when compared to satellite data. For this reason, in a previous publication (Pitarch et al., 2020; Table 3) we compared daily OC-CCI reflectances to Valente's database as well as to other two internal databases. Uncertainties were the lowest when OC-CCI daily products were compared to an internal database "CNR", collected using strict NASA protocols and uncertainties traceable to NIST standards. Relative RMS errors were within 20 % , with biases not significantly different from zero, except for the band at 412 nm. At 670 nm, errors were much higher in relative terms due to the low value of $R_{rs}(670)$, but they remained low in absolute units ($sr^{-1}$). These RMS errors include uncertainties present in the in-situ $R_{rs}$ as well, which may range between 5% to 10% (Zibordi et al., 2019), so an educated guess of a spectral average of 7.5 % was subtracted in quadrature from them to isolate the errors in the satellite data.

These remaining uncertainty estimates need to be propagated by the Secchi disk depth, hue angle and Forel-Ule algorithms. To assess this matter, three example $R_{rs}$ spectra were taken, representing ultra-oligotrophic, oligotrophic and eutrophic waters. Their respective Secchi disk depth, hue angle and Forel-Ule were calculated. Posteriorly, a Monte-Carlo analysis was made by adding the $R_{rs}$ uncertainties as Gaussian independent noise. This was repeated 10000 times and the histograms (Fig. 1) and related statistics (Table 2) of the resulting Secchi disk depth, hue angle and Forel-Ule were retrieved. It is shown that, for Secchi disk, biases have a spread of about σ= 22 % around the errorless value, with a negligible mean bias, increasing towards more turbid waters. The hue angle experiences a spread of about σ= 1 %, also with a negligible bias, though this result has to be interpreted keeping in mind the high average value and the low dynamic range of the hue angle. Statistics of the FU are much harder to interpret due to the strongly discrete nature of this variable, but essentially FU oscillates between the central value and the adjacent ones, in proportions that depend on the distances to these in hue angle terms.

A third uncertainty source is the remote algorithm itself with respect to the actual in-situ value. Since the algorithms were published elsewhere, this article does not validate them further. Here, we simply summarize statistical findings on their accuracy in each of the three next sub-sections.

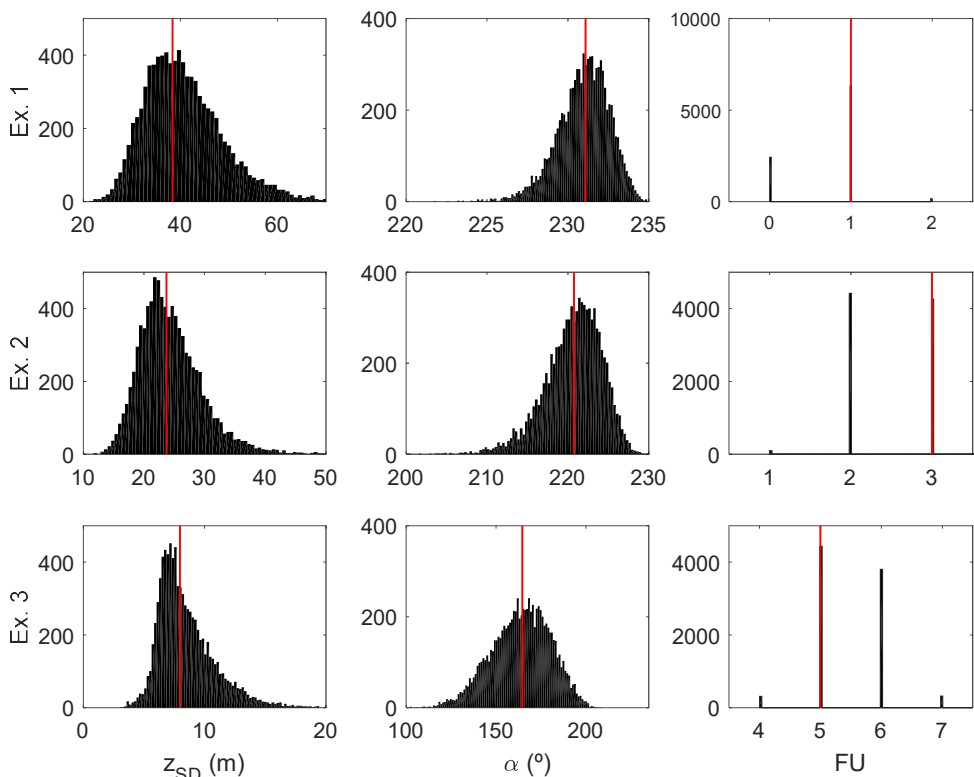

**Fig. 1 Monte Carlo simulation of error propagation from spectrum to Secchi disk depth, hue angle and Forel-Ule index. The upper panel shows the results of 10000 simulations (black bars) for ultra-oligotrophic waters, compared to the reference spectrum results (red line). The middle panels show the results for an oligotrophic water spectrum and the lower panels depict the results for a eutrophic water spectrum.**

175

**Table 2 Biases and standard deviation of the errors caused by random additive Gaussian perturbations to three given remote sensing reflectances from which the Secchi disk depth, the hue angle and the Forel-Ule index are derived. The "exact values" correspond to the red vertical lines in Fig. 1.**

| $z_{SD}$ (m) | | | A (º) | | | FU | | |
|---|---|---|---|---|---|---|---|---|
| Exact value | Bias (%) | σ (%) | Exact value | Bias (%) | σ (%) | Exact value | Bias (%) | σ (%) |
| 38.4 | 6.9 | 22.5 | 231 | -0.06 | 0.7 | 1 | -25.4% | 47.7% |
| 28.5 | 3.3 | 22.7 | 226 | -0.05 | 1.6 | 2 | -17.1% | 18.0% |
| 8.0 | 6.6 | 30.3 | 165 | -0.5 | 10.0 | 5 | 9.3% | 12.6% |

180





### 4.2. Secchi disk

The Secchi disk depth algorithm was validated by Lee et al. (2015). The $z_{sd}$ derived from in-situ $R_{rs}$ compared well to actual measurements with a Secchi disk, with a mean absolute percent difference (MAPD) of ~19 %. The linear regression slope between in-situ and remote estimation deviated marginally from the 1:1 line. Unfortunately, most of their data was not open to external users and the MAPD is a statistical parameter different from RMS or standard deviation. However, a quick numerical verification with Gaussian noise (not shown) reveals that the ratio between the relative RMS error and MAPD for unbiased estimates is very stable at about ~1.25. Therefore, Lee's MAPD=18.2 % equals to RMS=22.8 %. From this uncertainty, one must subtract the part due to the in-situ radiometry. Unfortunately, Lee's data, radiometric uncertainties are not traced, so a reasonable average 10 % RMS error is guessed here. Subtracted in quadrature from the 22.8 %, results in an uncertainty of ~20 % for Secchi disk model estimations respect to in-situ data. Overall, taking into consideration the uncertainties due to the satellite radiometry, we may conclude that Secchi disk depth derived from daily OC-CCI data has an uncertainty of ~32 % when compared to in-situ data.

### 4.3. Hue angle

Uncertainties in the hue angle product derive from the approximation of the spectral integrals involved in the algorithm from a limited set (6 here) of satellite bands (Van Der Woerd and Wernand, 2015;Van Der Woerd and Wernand, 2018;Pitarch et al., 2019b). These uncertainties vary with the water type because the few available spectral bands may be enough to capture all the optical information in simple case 1 waters, but not in optically complex waters. In fact, for case 1 waters, in relative terms, this amounts to less than 0.1 %, while in optically complex waters, it can amount up to a 3 %. We can therefore conclude that, for satellite hue angle estimations, the uncertainty is almost totally caused by the satellite radiometry, which ranges from less than 1 % to about 11 % as water shifts from very clear to turbid. This may also be an upper bound because we have assumed spectrally independent Gaussian noise at every satellite band, but errors of an atmospheric correction may have some spectral correlation. As a consequence of this, satellite-derived hue angle (and FU) seem to be particularly robust to uncertainties in satellite $R_{rs}$ after atmospheric correction (Wang et al., 2018).

### 4.4. Forel-Ule index

In-situ FU readings are different depending on whether a submerged Secchi disk is used to obtain a visual estimate of the colour over its surface or the estimate is obtained by looking at the water only. Essentially, an underwater white object reflects light that, after propagating across the water column back, is observed from above with a hue angle that is lower than that of the surrounding water. These differences are significant and the amount of this bias depends on the color of the water itself effect, which was previously shown after radiative transfer simulations (Pitarch, 2017), and recently shown experimentally (Nie et al., 2020). In addition, in-situ FU can be either estimated visually or derived mathematically from a $R_{rs}$ that has been measured in-situ. The equivalence of both must not be taken for granted and their relation has not been studied yet. In summary, there are several methodological concerns that affect the definition of reference FU measurements for comparison to satellite estimations.





In the interim, we choose to define reference FU measurements as those originating from in-situ hyperspectral $R_{rs}$, of which the hue angle has been calculated by integration and FU by discretization. In such case, the uncertainties are only those related to the satellite radiometry, as with the hue angle, with the overwhelming number of times, an error within ±1.

## 5. Global average maps

In order to provide the reader with a feeling of the global variability covered by these parameters, some basic statistics are demonstrated in this section. At every pixel the 256 values (21 years and 4 months of data) were combined and the arithmetic mean, standard deviation and coefficient of variation were calculated. Fig. 2 shows the results for the $z_{SD}$ time series. The mean map highlights the oligotrophic gyres, with the South-Pacific one reaching a mean of ~60 m in its most transparent core. The map shows

a global pattern that is similar when deriving other parameters like chlorophyll. In the oligotrophic gyres, the chlorophyll concentration is at minimum values (<0.1 mg m$^{-3}$) in agreement with the expected higher transparency induced by low concentration of phytoplankton cells, as also highlighted by the $z_{SD}$. The standard deviation of $z_{SD}$ shows interesting patterns. It is mostly driven by the seasonal variability and it is highest in the zones of the gyres that are furthest apart from the equator, showing the seasonal expanding

and shrinking of the subtropical gyres though they might also include the long-term effect of gyre expansion (Polovina et al., 2008). Other peaks of variability are zones affected by seasonal blooms like off Iceland or the Antarctica coast. This latter feature is most accentuated in the coefficient of variation.

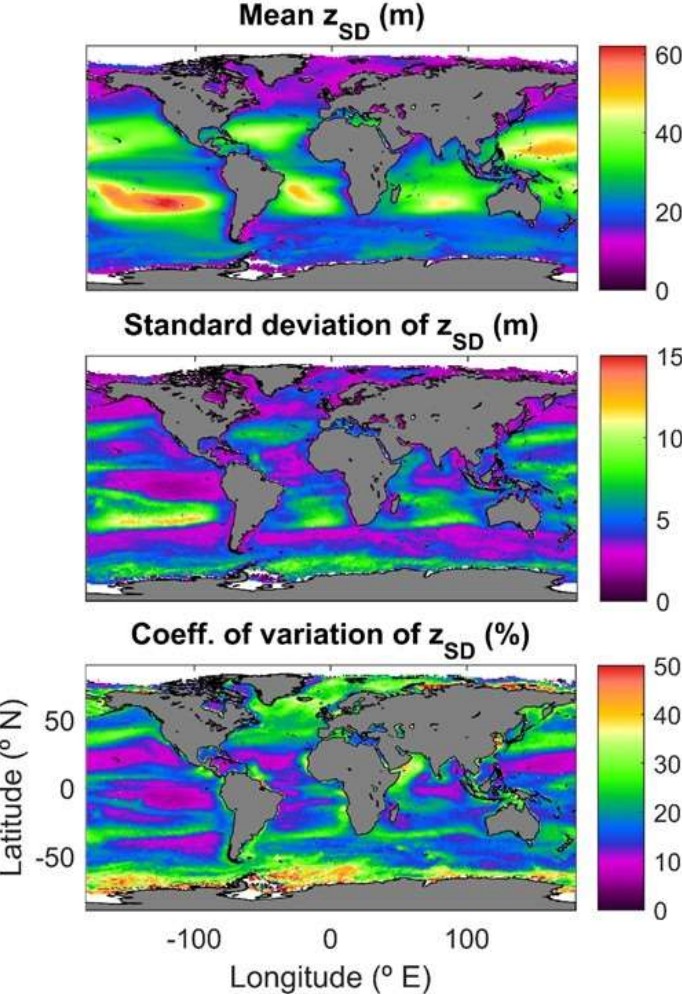

Figure 2: Global maps of the Secchi disk depth ($z_{SD}$) in the period 1997 - 2018. Top: series mean. Mid: standard deviation. Bottom: coefficient of variation.

The hue angle ($\alpha$) shows a pattern similar to $z_{SD}$ (Fig. 3), though with a marked non-linearity. Global values cover the range ~50º - 235º, though 85% of the world's oceans are blue with a hue angle above 200º. The standard deviation is very low in the oligotrophic areas, reflecting the fact that the human eye has little sensitivity to variations in color when these happen at the lower spectral end of the visible range. On the other hand, variability is mostly highlighted in zones of green waters like mesotrophic and eutrophic areas as well as turbid shelf seas and coastal zones.



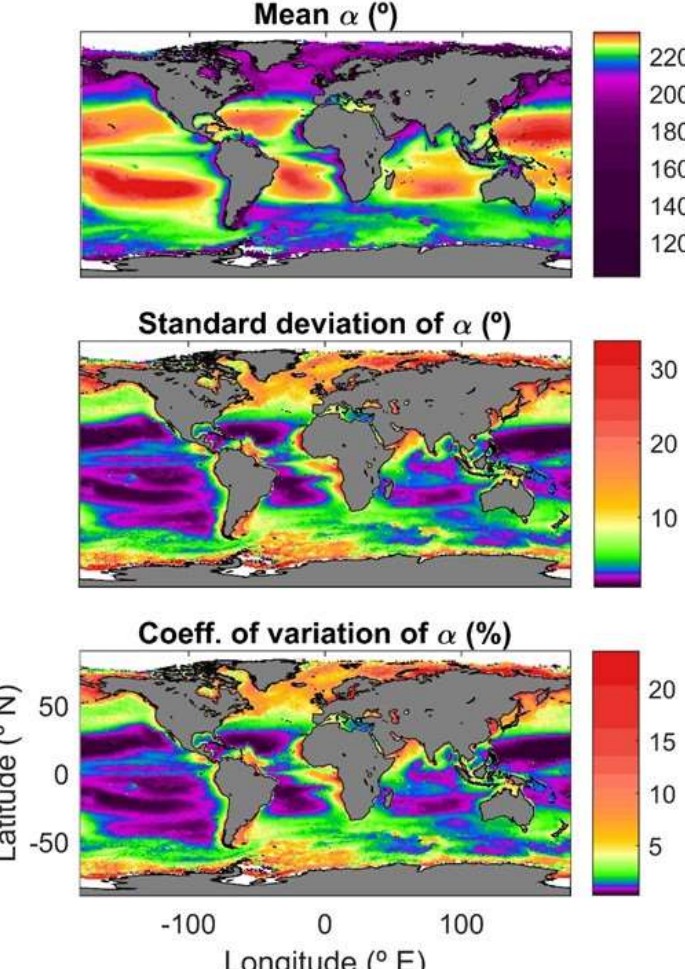

**Figure 3: Global maps of the hue angle (α) in the period 1997 - 2018. Top: series mean. Mid: standard deviation. Bottom: coefficient of variation.**

The Forel-Ule mean value (Fig. 4) reflects the known global patters. Despite being the scale from 0 to 21, the lowest 99 % of the data are within FU≤8, keeping the upper rest of the FU scale for very turbid coastal areas or highly humic waters. Note that, despite the fractional results due to statistics, FU are integers. As shown for the hue angle, FU shows very little variation for the oligotrophic and low latitude areas, reaching the widest dynamic range in coastal and high-latitude blooming seas.

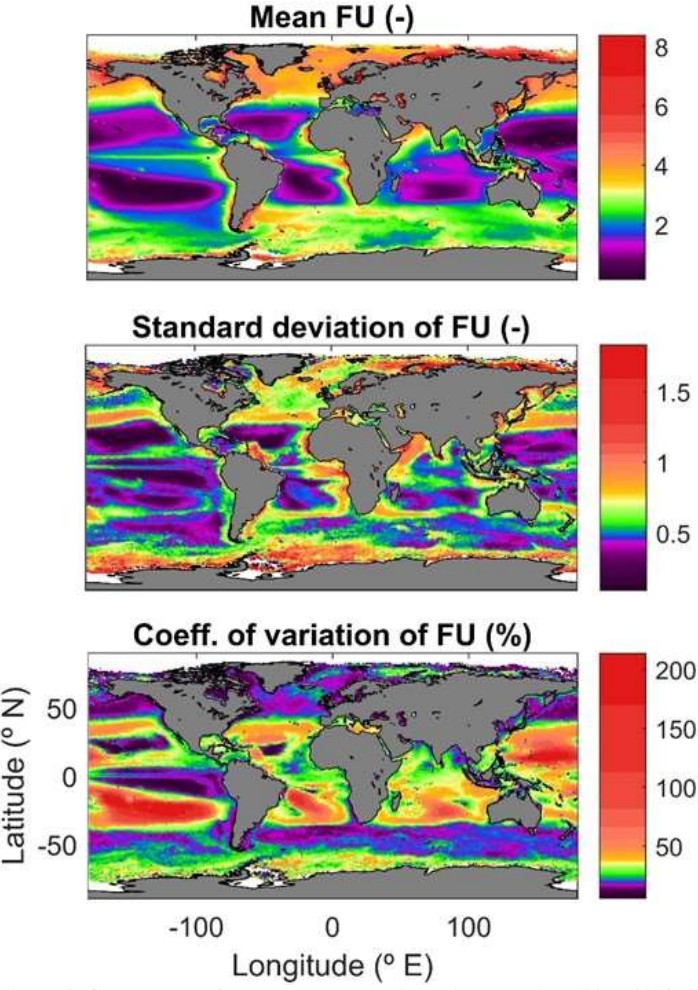

**Figure 4: Global maps of the Forel-Ule (FU) index in the period 1997 - 2018. Top: series mean. Mid: standard deviation. Bottom:**
**coefficient of variation.**

**6. Example of application: trend analysis for the North Atlantic Ocean**

As already argued, a main goal of the OC-CCI project is the construction of a time series for the
calculation of oceanic trends. Here, we used the lower spatial resolution product at 0.25°, covering the
265 Atlantic Ocean north of the equator. This section reports trend analysis of the area for the Secchi disk
depth and the hue angle. Seasonal decomposition was made pixel-wise with the publicly available BEAST

code (Zhao et al., 2019). The Sen's slope (Sen, 1968) of the de-seasonalized signal was calculated and the result was tested for significance at 95 % (Mann, 1945;Kendall, 1975). In Fig. 5 the upper two panels show only the statistically significant trend signals for $z_{SD}$ and α, calculated for the period 1997-2018. A

positive (negative) trend in $z_{SD}$ means that a Secchi disk will disappear from sight at a larger (shallower) depth, corresponding to a higher (lower) transparency. A positive (negative) trend in hue angle means that the spectrum shifts towards the blue (green), normally corresponding to a reduction (increase) in the chlorophyll concentration (Pitarch et al., 2019b). In the lowest panel of Fig. 5 the sign (positive or negative) of the trends are combined to yield 4 classes; a combination of positive $z_{SD}$ sign and negative

hue angle sign are given the code (+ -). Again, no colour indicates no significant trend in both indexes.

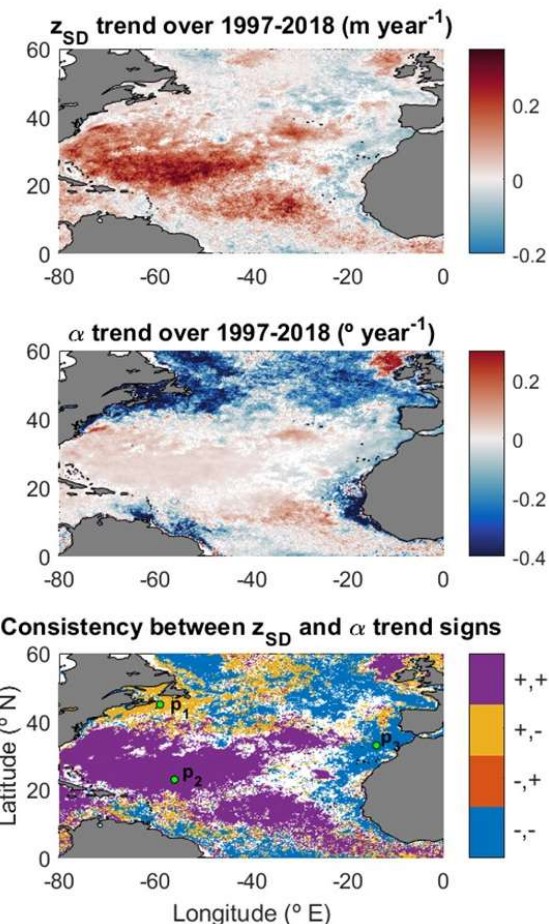

**Figure 5: Upper figure: linear slope of Secchi disk ($z_{SD}$) trend at pixel scale for the North Atlantic. Central figure: same as before, but for the Hue angle (α). Lower figure: signs of both linear slopes.**



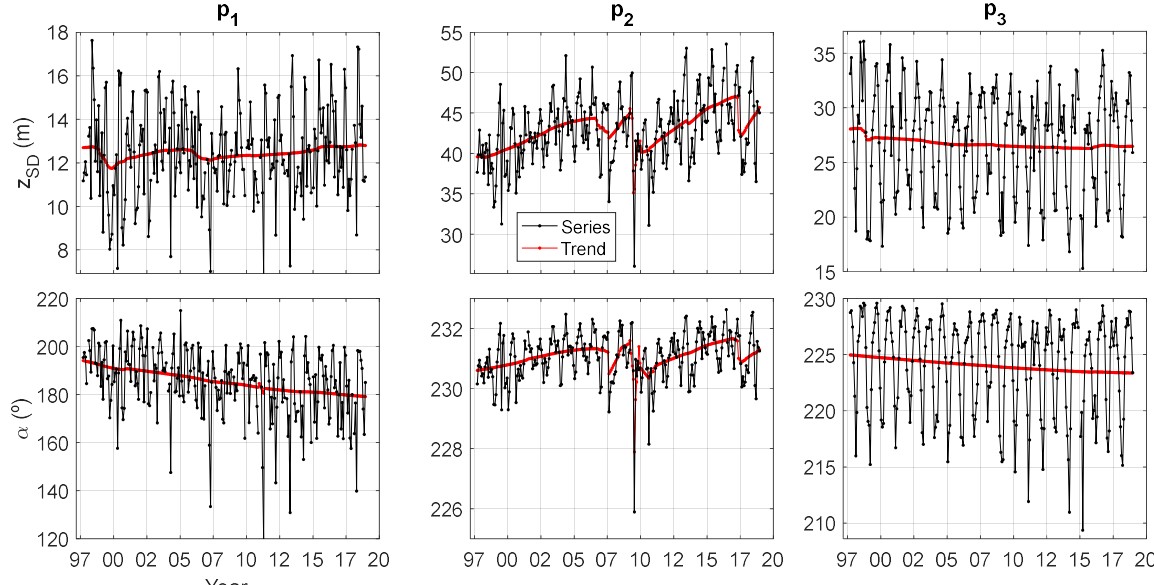

**Figure 6: Time series of Secchi disk depth (upper panels) and hue angle (lower panels) at three points in the North Atlantic Ocean. Red lines show the trend after de-seasonalization.**

The North-Atlantic trend maps (Fig. 5) show that the subtropical gyre is gaining transparency and becoming bluer. On the other hand, mesotrophic, more northern waters outside the gyre as well as coastal waters are experiencing totally different dynamics. In general, there is a loss of transparency and a green shift. Some areas where trends are very small have apparent opposite trends in the Secchi disk depth and hue angle, which highlights the fact that the Secchi disk depth and hue angle do not completely contain the same information, and it will deserve some future research. In Fig. 6, the complexity in the spatial and temporal behaviour is detailed for three points, their position indicated on the lower map of Fig 5. We anticipate that the study of these variation at various spatial scales and their relation to potential drivers like sea surface temperature (SST), nutrients, mixed layer depth coastal currents and river outflow will be explored.

### 7. Data availability

The 256-full resolution netCDF files constituting the global time series at monthly frequency and 4 km resolution is publicly available through the following link: https://doi.pangaea.de/10.1594/PANGAEA.904266 (Pitarch et al., 2019a). This also contains two teaser sub-products of much smaller size that can be quickly downloaded and easily handled locally: (1) a downgraded global dataset at 1 ° for latitude and longitude and (2) a downgraded dataset at 0.25 ° for latitude and longitude for the North Atlantic Ocean.



### 8. Conclusions


We have derived a consistently calibrated 21-year-long dataset of monthly Secchi disk depth, hue angle and Forel-Ule index over the global oceans, at ~ 4 km spatial resolution at the equator. The algorithms employed for this release have been recently published, are at the state of the art and their application is not restricted to any specific water type. Best-effort validation statistics have been provided based on current evidence. Global average maps have shown meaningful patterns that are consistent with the general understanding of the ocean dynamics. A small example on trend estimation has also been provided to illustrate one of the main areas of research that can be addressed with this dataset.


This dataset has been directly derived from the OC-CCI dataset, and as such, shares its advantages and limitations. Main advantages are the careful sensor inter-calibration to avoid spurious trends, the constant revision of the methodology in search for improved quality and the warranted continuation of the time series for the next years. Limitations are the exclusion of high latitude areas due to high solar zenith angle, that causes problems in the OC-CCI processing and the uneven monthly binning depending on cloud cover. These issues could be solved in a future release, which could start from daily data with an application of a gap-filling procedure.


With this release, we hope to eliminate redundant efforts within the community to estimate water transparency and colour from satellite measurements. Depending on the reception and user needs, future releases using other temporal and spatial resolutions could be made.


For $z_{SD}$ and FU, these time series could be further enlarged backwards by including the archived in-situ data that go back to the year 1890. However, even in the absence of historic data, twenty-one years of satellite data are already a valuable dataset that may provide new insights on the ocean's variability in relation to climate change and constitute valuable information to train and validate mechanistic oceanic models.


### 9. Author contribution

J.P. conceived the manuscript, produced the dataset and associated results and wrote the first draft of the manuscript, following advice by M.B. and S.M. M.B., H.vdW. and S.M. revisions of the draft. J.P. wrote the final submitted version.


### 10. Competing interests

The authors declare that they have no conflict of interest.

### 11. Acknowledgements


J.P. thanks financial support by the Ministry for Science and Culture of Lower Saxony, Germany in the framework of the 'Coastal Ocean Darkening' project (VWZN3175) and by the Copernicus Climate Change Service (C3S_511). M. B. is supported by the ESA Living Planet Fellowship Project



PHYSIOGLOB: Assessing the inter-annual physiological response of phytoplankton to global warming using long-term satellite observations, 2018-2020.

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
