# Peer review of "Global maps of Forel-Ule index, hue angle and Secchi disk depth derived from twenty-one years of monthly ESA-OC-CCI data"

_Earth System Science Data, 2020_

## Referee Comment (RC1) · Anonymous Referee #1 · 1 Dec 2020

The authors present a publicly available, monthly, global dataset of the Forel-Ule index, hue angle, and Secchi disk depth derived from the ESA Ocean Colour Climate Change Initiative satellite times series of spectral remote-sensing reflectances. Generally speaking, the manuscript is well written and straightforward – almost terse, which I find acceptable as the algorithms and approaches are well documented. Overall, I see no major show-stoppers in its publication. Minor comments are provided below, most of which focus on grammar and presentation.

Line 38: Suggest expanding satellite acronyms, adding agencies, and adding mission timelines, e.g., "the NASA Moderate Resolution Imaging Spectroradiometer onboard

[Figure]

Aqua (Aqua-MODIS; 2002-present)" – or, changing "present" in the former to the last year included in OC-CCI.

L42: Please indicate spectral dependency and provide units the first time a variable is introduced.

L72: Remove out of place "Rrs" or revise to "calculate them from Rrs".

L80: "Downloaded product is the" is grammatically awkward. Do you mean "The source product for all derived variables presented in this dataset . . .".

L99: Expand CIE acronym.

General comment on uncertainties: Just confirming that temporal, pixel-level uncertainties are not possible? This is generally apparent from this section, but would still be useful for clarity to state as such up front.

L135: Suggest mentioning that any systematic bias in the geophysical products will not be reduced like random noise.

L154: What is the remaining uncertainty estimate after 7.5% was subtracted in quadrature? Or, was the subtraction included in the "within 20%" stated on line 149? Even if reported elsewhere, the spectral values of calculated Rrs uncertainties (used on line 57) are worth repeating here. What other terms are included in the (quadrature-expressed) uncertainty budget?

Line 252: Global "patterns". And, suggest rewording to "Despite the scale ranging from 0 to 21".

Line 264: Neither the 0.25-deg nor the 1-deg datasets have been described yet. Generation of these datasets needs to be presented earlier in the manuscript.

---

## Referee Comment (RC2) · Anonymous Referee #2 · 17 Dec 2020

This study provides a new, open access dataset that consists of global maps of the Forel-Ule index, hue angle and Secchi disk depth and can be conveniently downloaded from PANGAEA. A merged multi-sensor data (OC-CCI) was used as the source data and the algorithms can be traced from other documents. Generally, this is meaningful work and facilitates the research of other scientists in the water color remote sensing community. I suggest making the following minor revisions before the publication of this

study:

L26: "easier to handle" should be "easier-to-handle"

L49: delete "the" before "water surface", delete "is" before "tracked"

L50: delete "," before "(Wernand, 2010)"

L72: better to change "so far" to "thus far", "them" should be "their"

L95: delete "the" before "deep bule"

L133: As daily OC-CCI products are also available and can be used to match with in-situ data, why are daily variables not included in this new dataset? Or as an alternative, if possible, the authors could publish their code on GitHub, perhaps a function that makes Rrs the input and FU index and other variables the output.

L143: delete "and" before "without"

L155: The minimum "exact" z_SD in these three experiments is set to 8.0 m, which limits the verification to case 1 waters. But it is obvious that the nearshore seawater will be much more turbid; therefore, is the dataset provided by this research still reliable in turbid water (for example FU>10)?

Table 2: The "exact value" of FU index in EX.2 does not match that in Fig. 1, please check it. And please change "A(°)" to "$\alpha$(°)" to keep it consistent with Fig. 1.

L189: "RMS=22.8%", do you mean "relative RMS=22.8%"?

L195: How is this "∼32%" calculated, "22.8%"+"10%"?

L290: "variation" should be "variations"

---

## Author Comment (AC1) · 8 Jan 2021

Reply to anonymous Referee #1

The authors present a publicly available, monthly, global dataset of the Forel-Ule index, hue angle, and Secchi disk depth derived from the ESA Ocean Colour Climate Change Initiative satellite times series of spectral remote-sensing reflectances. Generally speaking, the manuscript is well written and straightforward – almost terse, which I find acceptable as the algorithms and approaches are well documented. Overall, I see no major show-stoppers in its publication. Minor comments are provided below, most of which focus on grammar and presentation.

We appreciate the positive opinion of the referee and thank him/her very much for the time and feedback to improve the manuscript. We have taken all comments in consideration and have improved the manuscript accordingly. We agree on the "terse style", but unfortunately ESSD articles can be like this, as the journal forbids any scientific interpretation of the data. Some minor editing has been made here and there to make parts smoother to read. With the example of the North Atlantic Ocean, we intended to provide some hints on further use and make the data more interesting for scientists.

Line 38: Suggest expanding satellite acronyms, adding agencies, and adding mission timelines, e.g., "the NASA Moderate Resolution Imaging Spectroradiometer onboard Aqua (Aqua-MODIS; 2002-present)" – or, changing "present" in the former to the last year included in OC-CCI.

We have expanded the sensors' names, though not been explicit on the carrying platforms, in order to not to make the text cumbersome.

L42: Please indicate spectral dependency and provide units the first time a variable is introduced.

Done.

L72: Remove out of place "Rrs" or revise to "calculate them from Rrs".

Corrected.

L80: "Downloaded product is the" is grammatically awkward. Do you mean "The source product for all derived variables presented in this dataset...".

Indeed, this is much better. Corrected.

L99: Expand CIE acronym.

Done.

General comment on uncertainties: Just confirming that temporal, pixel-level uncertainties are not possible? This is generally apparent from this section, but would still be useful for clarity to state as such up front.

Indeed, the estimation of pixel-level uncertainties is not possible for the products released in this dataset. We are aware of this lack, but such information can only be given after a "spectral class discrimination", using a large and quality-controlled in-situ dataset that spans the broadest dynamic range. In fact, such stringent requirements have the effect that not all products can be released with pixel-wise uncertainties in the OC-CCI dataset (Jackson et al., 2019). We have elaborated on this point in section 4. Validation in the revised version.

L135: Suggest mentioning that any systematic bias in the geophysical products will not be reduced like random noise.

Done.

L154: What is the remaining uncertainty estimate after 7.5% was subtracted in quadrature?  Or, was the subtraction included in the "within 20%" stated on line 149? Even if reported elsewhere, the spectral values of calculated Rrs uncertainties (used on line 57) are worth repeating here. What other terms are included in the (quadrature-expressed) uncertainty budget?

The 7.5 % is a reasonable uncertainty estimate of the in-situ Rrs measurements. The statistics "within 20 %" were taken from Table 3 of Pitarch et al. (2020) where we made a matchup between in-situ Rrs and OC-CCI Rrs. Subtraction in quadrature of 7.5 % from these Rrs uncertainties are the input to the Monte Carlo simulations of the hue angle, Forel-Ule and the Secchi disk depth whose results we report in Figure 1 and Table 2 of the present paper. All in all, we think that we have provided sufficient information for readers to follow and even duplicate out Monte Carlo experiment. Note that, following the suggestion by Reviewer 2, the code has been made publicly available.

Line 252: Global "patterns". And, suggest rewording to "Despite the scale ranging from 0 to 21".

Corrected.

Line 264: Neither the 0.25-deg nor the 1-deg datasets have been described yet. Generation of these datasets needs to be presented earlier in the manuscript.

These datasets are now introduced in section "3. Product description".

References

Jackson, T., Chuprin, A., Sathyendranath, S., Grant, M., Zühlke, M., Dingle, J., Storm, T., Boettcher, M., and Fomferra, N.: Ocean Colour Climate Change Initiative (OC_CCI) –Interim Phase. Product User Guide. D3.4 PUG, 2019.

Pitarch, J., Bellacicco, M., Organelli, E., Volpe, G., Colella, S., Vellucci, V., and Marullo, S.: Retrieval of Particulate Backscattering Using Field and Satellite Radiometry: Assessment of the QAA Algorithm, Remote Sensing, 12, 77, https://doi.org/10.3390/rs12010077, 2020.

---

## Author Comment (AC2) · 8 Jan 2021

This study provides a new, open access dataset that consists of global maps of the Forel-Ule index, hue angle and Secchi disk depth and can be conveniently downloaded from PANGAEA. A merged multi-sensor data (OC-CCI) was used as the source data and the algorithms can be traced from other documents. Generally, this is meaningful work and facilitates the research of other scientists in the water color remote sensing community. I suggest making the following minor revisions before the publication of this study:

We greatly appreciate the reviewer's positive view on this article and acknowledge the careful reading and the useful suggestions to improve the text. We have made the following changes:

L26: "easier to handle" should be "easier-to-handle"

Corrected.

L49: delete "the" before "water surface"

We believe the presence of the "the" is necessary, because the action of lowering is with respect to the surface.

delete "is" before "tracked"

Corrected.

L50: delete "," before "(Wernand, 2010)"

Corrected.

L72: better to change "so far" to "thus far"

Corrected.

"them" should be "their"

We believe "them" is the appropriate word here, because it does not refer to persons.

L95: delete "the" before "deep blue"

Corrected.

L133: As daily OC-CCI products are also available and can be used to match with in-situ data, why are daily variables not included in this new dataset? Or as an alternative, if possible, the authors could publish their code on GitHub, perhaps a function that makes Rrs the input and FU index and other variables the output.

Thank you, this is a great suggestion. We have uploaded the code that calculates the hue angle, Forel-Ule index and Secchi disk depth from Rrs to a GitLab repository, with the appropriate link given in the article.

We focused here on monthly data because that is the time resolution commonly used in satellite climate studies. Daily data will be much more demanding in terms of processing and storage, but we may consider such a release if the present dataset has a good reception.

L143: delete "and" before "without"

Corrected.

L155: The minimum "exact" $z_{SD}$ in these three experiments is set to 8.0 m, which limits the verification to case 1 waters. But it is obvious that the nearshore seawater will be much more turbid; therefore, is the dataset provided by this research still reliable in turbid water (for example FU>10)?

Indeed, this algorithm is applicable to waters with $z_{SD}$ values of less than 1 m, as is evidenced by the good match between the in-situ $z_{SD}$ and the Rrs-derived $z_{SD}$; see Lee et al. (2015). The choice of a particular spectrum that leads to $z_{SD}$ =8.0 m is just one arbitrary example among three to illustrate the effect of uncertainty in Rrs on the derived $z_{SD}$ value.

Table 2: The "exact value" of FU index in EX.2 does not match that in Fig. 1, please check it. And please change "A(◦)" to "α(◦)" to keep it consistent with Fig. 1.

Corrected.

L189: "RMS=22.8%", do you mean "relative RMS=22.8%"?

Yes, these are relative units.

L195: How is this "~32%" calculated, "22.8%"+"10%"?

Uncertainties are added in quadrature. This is now explicitly mentioned in sections 4.1 and 4.2 .

L290: "variation" should be "variations"

Corrected.

---

## Author Response (AR2)

Dear Editor,

We thank you for the fine screening of the manuscript. We have revised it and we are happy to resubmit it according to your comments, that are addressed below. Other than these, a rephrased figure caption and an additional note in the acknowledgements, there are no further changes.

Best regards,

Jaime Pitarch.

Author's reply to:

Topical Editor Decision: Publish subject to minor revisions (review by editor) (13 Jan 2021) by Giuseppe M.R. Manzella

Comments to the Author:

Line 24 of the abstract: the link to data is https://doi:10.1594/PANGAEA.904266. Must be https://doi.pangaea.de/10.1594/PANGAEA.904266 (please check)

Thank you for finding this error. It is corrected now.

Line 154 - 155: As the situation for the variables treated in this article is similar in terms of paucity of publicly of quality-controlled available data ... Please rephrase the sentence.

We appreciate the suggestion and acknowledge that the phrasing was odd. Now it reads "As the satellite variables derived in this article are also affected by a lack of open and quality-controlled in-situ data, uncertainties will be provided as bulk estimates."

Line 282 - oceanic trends and oscillations. ?oceanic trends and seasonal variability? Please specify.

We agree that this sentence needed revision. Now it reads "This section provides an example of application that shows the unique possibilities of long-term satellite data: the calculation of pixel-wise oceanic trends."